# Quantitative and Predictive Modelling of Lipid Oxidation in Mayonnaise

**DOI:** 10.3390/antiox10020287

**Published:** 2021-02-15

**Authors:** Donny W. H. Merkx, Andries Swager, Ewoud J. J. van Velzen, John P. M. van Duynhoven, Marie Hennebelle

**Affiliations:** 1Unilever Food Innovation Centre, Bronland 14, 6708 WH Wageningen, The Netherlands; Donny.Merkx@unilever.com (D.W.H.M.); Ewoud-van.Velzen@unilever.com (E.J.J.v.V.); 2Laboratory of Food Chemistry, Wageningen University & Research, Bornse Weilanden 9, 6708 WG Wageningen, The Netherlands; andriesswager@gmail.com (A.S.); Marie.Hennebelle@wur.nl (M.H.); 3Laboratory of Biophysics, Wageningen University & Research, Stippeneng 4, 6708 WE Wageningen, The Netherlands

**Keywords:** hydroperoxides, aldehydes, on-set time, oil-in-water emulsion, accelerated shelf-life, ^1^H NMR, quantification, Foubert model

## Abstract

Food emulsions with high amounts of unsaturated fats, such as mayonnaise, are prone to lipid oxidation. In the food industry, typically accelerated shelf life tests are applied to assess the oxidative stability of different formulations. Here, the appearance of aldehydes at the so-called onset time, typically weeks, is considered a measure for oxidative stability of food emulsions, such as mayonnaise. To enable earlier assessment of compromised shelf-life, a predictive model for volatile off-flavor generation is developed. The model is based on the formation kinetics of hydroperoxides, which are early oxidation products and precursors of volatile aldehydes, responsible for off-flavor. Under accelerated shelf-life conditions (50 °C), hydroperoxide (LOOH) concentration over time shows a sigmoidal curvature followed by an acceleration phase that occurs at a LOOH-concentration between 38–50 mmol/kg, here interpreted as a critical LOOH concentration (CC_LOOH_). We hypothesize that the time at which CC_LOOH_ was reached is related to the onset of aldehyde generation and that the characterization of the LOOH-generation curvature could be based on reaction kinetics in the first days. These hypotheses are tested using semi-empirical models to describe the autocatalytic character of hydroperoxide formation in combination with the CC_LOOH_. The Foubert function is selected as best describing the LOOH-curvature and is hence used to accurately predict onset of aldehyde generation, in most cases within several days of shelf-life. Furthermore, we find that the defining parameters of this model could be used to recognize antioxidant mechanisms at play.

## 1. Introduction

In high-fat food emulsions such as mayonnaise, lipid oxidation is a major concern, due to the high amount of unsaturated lipids that can undergo oxidation reactions. The first step is a free radical reaction that yields hydroperoxides (LOOHs), a meta-stable product that can degrade further into aldehydes, epoxides and other secondary oxidation products. Antioxidants (AOs) can be used to combat this process, heavily depending on the type of the oil matrix, where, for example fish oil [1], oat flakes [2] and mayonnaises [3,4] all have different optimal AO systems. To assess these AOs and thus the oxidative stability of high-fat food emulsions, which under consumer relevant conditions are stable for more than a few months, the food industry applies accelerated shelf-life tests. Here the oxidation reactions are stimulated to enable faster assessment [5,6], which can be achieved in several ways, such as the addition of pro-oxidants like transition metals [3,7,8], manipulation of oxygen content and state [9], or most commonly, the increase of storage temperature [10,11]. In these enhanced storage tests, different markers can be measured over time. One of the most frequently monitored markers is hexanal, a secondary lipid oxidation product that is linked to the perceived off-taste and off-flavor of the product. However, even under accelerated conditions, hexanal assessment still requires several (4–8) weeks to be assessed accurately. Earlier markers are the hydroperoxides, being the first stable oxidation products to be formed, these can be detected from the first days onward. However, not these hydroperoxides but the aldehydes are responsible for the off-taste and smell of the final product. To this end, it would be of great value to quantitatively predict the onset of aldehyde generation using initial hydroperoxide generation.

In this work, lipid oxidation in food emulsions is accelerated by storage at elevated temperature (50 °C), where literature suggests that inter-droplet transport and diffusion mechanisms are not rate limiting [12,13]. We hypothesize that the main mechanisms at play are the known free radical mediated reactions and we also attribute a colloidal role to reverse micelle formation, which has been shown to play a part in lipid oxidation kinetics [14,15]. First, these free radical reactions are autocatalytic reactions that can be observed as sigmoidal trends [16,17,18]. In this autocatalytic reaction, the initial phase can be hampered by the presence of primary antioxidants or the general lack of initiators. This is followed by a sudden increase of oxidation products being formed, due to the ample presence of the unsaturated double bonds and oxygen. The generation of hydroperoxides will then decelerate, due to the consumption of oxygen, the decrease of oxidizable substrate and the degradation of the hydroperoxides, resulting in sigmoidal time-response curves. Sigmoidal time-response curves can be described mathematically by exponential functions with three or four parameters, as described in Gompertz [19] and Foubert, Dewettinck [20]. Second, reverse micelles might play a critical role in the propagation of lipid oxidation products, as they would be able to capture pro-oxidant water soluble compounds and transport them throughout the lipid layer [21,22]. Micellization is reported to be stimulated even more by the accumulation of lipid oxidation products, specifically the hydroperoxides [23]. In other words, after a critical concentration of hydroperoxides (CC_LOOH_) formed at an oxidizable interface, these hydroperoxides will stabilize reverse micelles and promote further oxidation inside the droplet. We hypothesize that when the CC_LOOH_ is reached, both primary and secondary oxidation mechanisms are enhanced significantly, making the rapid acceleration of aldehydes imminent. In recent years, new methods, utilizing Liquid Chromatography-Mass Spectrometry (LC-MS) and Electron Spin Resonance (ESR) [24,25] have been developed to complement the existing methods, such as hexanal by Gas Chromatography (GC) and Peroxide Value (PV) [26,27] that allow fast and quantitative measurements of the key primary and secondary oxidation markers. Moreover, proton Nuclear Magnetic Resonance (^1^H NMR) methods have been developed that provide a rapid and accurate quantification of the bulk of hydroperoxides and (non-volatile) aldehydes [28]. The simultaneous quantification of both primary and secondary lipid oxidation products raises new opportunities to gain more mechanistical insight into the process and to develop quantitative predictive modelling approaches.

Here, we assess which of the sigmoidal Gompertz and Foubert functions best describe best hydroperoxide generation (measured by ^1^H NMR) in mayonnaise in accelerated shelf-life tests. Next, the selected function is used to predict the aldehyde onset, on the basis of the hypothesis that reaching the CC_LOOH_ is a trigger for accelerated secondary oxidation. This work was initially done on a wide set of data collected throughout the years, with a wide variety of ingredients and formulations. Afterwards, we validate this semi-empirical prediction model, using a designed experiment which includes antioxidants with known mechanisms. The fitted model parameters are interpreted in mechanistic terms.

## 2. Material and Methods

### 2.1. Raw Materials

CDCl_3_, DMSO-d6 and 4Å molsieves were purchased from Euriso-top (Saint-Aubin, France). Rapeseed (RP) oil, egg yolk, egg blend, sodium chloride, and spirit vinegar were purchased from local suppliers. Ethylenediaminetetraacetic acid calcium disodium salt (CaNa_2_EDTA) and gallic acid were purchased from Sigma Aldrich (Zwijndrecht, the Netherlands).

### 2.2. Sample Preparation

For the model selection and initial prediction, a wide range of natural and chemical antioxidants in different concentrations (more details in Section 2.4 and Appendix A) were used. Emulsification was done using emulsification with a Silverson mixer (Papendrecht, the Netherlands) for 6 min at 8.500 rpm. The water phase (final concentration 22%–35% *w/w*) consisted mainly of egg yolk (~5% *w/w*) or egg blend (~8% *w/w*) and NaCl (1%–1.5% *w/w*). Prior to the emulsification, antioxidants (if dosed) were added to the water phase. Next, rapeseed oil (65%–78% *w/w*) was slowly added to form the emulsion and finally spirit vinegar (1.4% *w/w*) to achieve the desired pH (3.8). In the validation study (Section 3.4 and Section 3.5), mayonnaises were prepared with egg yolk (5 % *w/w*) and rapeseed oil (78%). Ethylenediaminetetraacetic acid (EDTA/E) and Gallic Acid (GA/G) were added via the water phase. To obtain the different EDTA levels, EDTA-stock solution of 0.4% *w/w* in H2O replaced part of the water phase: respectively, 0.25 g, 0.50 g, and 1.88 g to obtain 10, 20, and 75 ppm EDTA. For the different GA concentrations, a GA-stock solution of 1% *w/w* in H2O was added: respectively 3.0 g and 6.0 g to obtain 75 and 150 ppm GA. The samples 75E-0G and 0E-0G and the center sample 10E-75G were prepared in duplicate. For the accelerated shelf-life tests (ASLTs), two aliquots of 1 g mayonnaise per formulation were stored in 20 mL screwcap vials in the dark at 50 °C for shelf-life testing. Upon collection, the aliquots were then stored at −20 °C until further analysis.

### 2.3. NMR Measurements

For the quantification of hydroperoxides and aldehydes in these aliquots, the oil phase was first obtained by freeze-thawing the mayonnaise to break the emulsion followed by centrifugation (VWR, MicroStar 17R, the Netherlands) for 5 min at 17,000× g to obtain a good separation of the oil and water layers. Of the oil phase, 150 µL were dissolved in 450 µL 5:1 CDCl_3_:DMSO-d6 solvent. The mixture (600 µL) was transferred to 5-mm NMR tubes. Single pulse and band selective ^1^H NMR spectra were then recorded on a 600 MHz (14.1 T) Bruker Avance III NMR spectrometer (Bruker BioSpin, Switzerland). Details on the assignment of signals and quantification can be found in Merkx, Hong [28].

### 2.4. Model Selection and Prediction Data

The dataset that was used for the model selection was pooled from a collection of trials performed in-house, with 238 formulations in total. Out of this data, two groups were defined. A schematic overview of the formulations used in this study can be found in the Appendix A. One overarching group of datasets (Set D in Appendix A) contained all data that had LOOH-curves at 50 °C, with at least five time points measured in duplicate, that were used for the model selection steps. Of this overarching group, a subgroup consisting of 23 samples from three different trials (Set A, B, and C in Appendix A) were stored for more than 30 days and had complementary aldehyde data to be used for the prediction studies. First, set D was used for selecting the sigmoidal model that provides the best fit to the LOOH-curves under ASLT (Section 2.4.1 and Section 2.4.2). Next, data from sets A, B, and C was used to build a model for predicting the aldehyde onset (Section 2.4.3).

#### 2.4.1. Sigmoid Models

For the description of the LOOH generation, two sigmoidal functions were explored: the Gompertz [19] and Foubert [29] functions, described in Equations 1 and 2, respectively:(1)$$C_{\mathrm{Ox}}{=\;A\;\times \;e}^{-e^{K(t-t0)}}$$
(2)$$C_{\mathrm{Ox}}=-A\;\times \;{\left[1+\left({\left(\frac{A-f0}{A}\right)}^{1-n}-1\right){\times e}^{-\left(1-n\right)\;\times \;K\;\times \;t}\right]}^{\frac{1}{1-n}}-A$$
where $C_{\mathrm{Ox}}$ is the concentration of oxidation product (mmol/kg) and t the storage time (day). Parameter A indicates the upper asymptote (mmol/kg) and parameter K the growth rate (day^-1^). For Equation 1 (Gompertz), the parameter t0 is the inflection time in days of the curve (day). In Equation 2 (Foubert), this inflection time is rewritten as f0, which can be interpreted as the initially present oxidation products (mmol/kg). Parameter n is a constant related to the asymmetry of the curve. The regression was done using the Excel Solver add-in using GRG Non Linear fitting with the boundaries as described in Appendix A.

#### 2.4.2. Model Selection Criteria

To find the best model for describing hydroperoxide generation, three information criteria were evaluated. These three criteria were the Aikake Information Criterion (AIC), Bayesian Information Criterion (BIC) and Law of Iterated Logarithm Criterion (LILC) [30]. For every curve, the AIC, BIC and LILC values were collected. Here, the model with the lowest criterion-value was considered the best model. An F-test was performed to evaluate in how many cases the full Foubert model (F4, including four variable parameters) was significantly better than the reduced Foubert model (F3, three variable parameters and one fixed parameter (n)).

#### 2.4.3. Prediction Model

In the prediction model of the aldehyde onset time, the regressor (t) was set at the time the CC_LOOH_ reached a concentration of 38 mmol/kg (Equation (3), rewritten form of Equation (2)). The predictor was the onset time of the aldehydes. The objective cross-point determination was adapted and slightly modified from Pinchuk and Lichtenberg [17]. A schematic overview of this procedure is shown in the Appendix A. In brief, the aldehyde curvature was first described using regression with the reduced Foubert function using *n* = 1.1 (best model to describe aldehyde generation from the model selection; data not shown). From this, its derivative function (Equation (4)) was used to estimate the maximum slope (Sx) using the Excel Solver add-in. The intercept of this slope with the baseline, set at 1.0 mmol/kg, was interpreted as the onset time of the aldehydes. The prediction was performed using linear regression without transformation in the form of y = ax.
(3)$$t=\ln\left(\frac{{\left(\frac{-C_{\mathrm{LOOH}}+A}{A}\right)}^{\frac{1-n}{1}}-1}{{\left(\frac{A-f0}{A}\right)}^{1-n}-1}\right)\times \frac{1}{\left(n-1\right)\cdot K}$$
(4)$$S_{x}=A\left({\left(\frac{A-f0}{A}\right)}^{1-n}-1\right){\times \mathrm{Ke}}^{-\left(1-n\right)\mathrm{Kx}}{\left(\left({\left(\frac{A-f0}{A}\right)}^{1-n}-1\right)e^{-\left(1-n\right)\mathrm{Kx}}+1\right)}^{\frac{1}{1-n}-1}$$
where t is the timepoint at which the hydroperoxide concentration $C_{\mathrm{LOOH}}$ is reached (day). $S_{x}$ is the slope (first derivative) of Equation (2) (mmol/kg/day). Parameter A is the upper asymptote (mmol/kg), K the curvature constant (day^−1^), f0 the initially present oxidation products (mmol/kg) related to induction time, n the asymmetry constant of the Foubert function.

### 2.5. Model Validation and Chemical Parameter Interpretation

A nonlinear least-squares curve-fitting procedure (lsqcurvefit) in Matlab R2020a (The MathWorks, Inc.) was used for estimating the model parameters A, K, and f0 in the reduced Foubert function (Equation (2)). The standard error for each model parameter was determined via a bootstrap regression procedure with residual resampling [31] and included 1000 repetitions. As the statistical distribution of the model parameters was unknown, the (non-parametric) bootstrap was considered as the most appropriate approach in our error estimate. A two-way ANCOVA (ANalysis of COVAriance) was performed to establish whether the contribution of different AOs on parameters K and f0 were statistically significant (at α = 0.05). Prior to the ANCOVA, both K and f0 were log-transformed to adjust for non-normality in the data. The regression coefficients of the ANCOVA model were used to determine the degree of association between the AOs and the dependent variables K and f0 (Equations (5) and (6)). Since the AOs were in the same concentration regime, the absolute size of these model coefficients could directly be interpreted as (relative) effect sizes. These analyses were done with log transformation of f0 and K in R (R Core Team, 2020) and the lm function in the stats (v3.6.2) package.
(5)$${K=\;\mu +\;\alpha \mathrm{AO}}_{i}{+\;\beta \mathrm{AO}}_{j}{+\;\gamma \mathrm{AO}}_{i}{\mathrm{AO}}_{j}+\;\varepsilon$$
(6)$${f0=\;\mu +\;\alpha \mathrm{AO}}_{i}{+\;\beta \mathrm{AO}}_{j}{+\;\gamma \mathrm{AO}}_{i}{\mathrm{AO}}_{j}+\;\varepsilon$$
where K and f0 are the dependent variables (Equation (2)), μ the mean, α, β, and γ the regression coefficients (ppb^−1^) of the corresponding term and ε the residual error. AO stands for the antioxidant (ppb), with the indexes i and j. for the type of AO.

## 3. Results and Discussion

### 3.1. Justification of Sigmoidal Models to Describe Hydroperoxide Formation

Typical curves of hydroperoxide generation during an accelerated shelf-life test (ASLT) are shown in Figure 1A. The three curves represent mayonnaises prepared with different antioxidant (AO) levels and follow a sigmoidal curvature up to a concentration of around 38 mmol/kg. From here onwards, the LOOH generation enters a second acceleration phase. The initiation of this secondary acceleration is believed to be related to the generation of aldehydes. This secondary acceleration can be explained by the recently introduced theoretical framework [12,13], which suggests that at certain concentrations, surface active hydroperoxides from inverse micelles propagate the oxidation reaction in a quicker rate. This aligns with our observations, and we therefore refer to this concentration when this secondary acceleration is observed (38 mmol/kg) as the Critical Concentration of LOOHs (CC_LOOH_).

To quantitatively describe these sigmoidal initial phases of LOOH generation (up to the CC_LOOH_), two model functions were explored: the Gompertz [19] and the recently developed Foubert [29] function. The Foubert function is an adaptation of the Gompertz function that includes a parameter for asymmetry. The inclusion of an asymmetry parameter allows for a better description of processes that have competing reactions. Foubert et al. developed this to describe crystal growth, where, besides growth and sigmoid progression, they also needed to consider the reverse reaction. In lipid oxidation, the degradation of LOOHs can similarly reduce the symmetry of the sigmoidal curve. Furthermore, other parameters that describe the shape of the sigmoidal curve, such as induction time (t0/f0) and growth rate (K), can hold information regarding the oxidative mechanisms at play and will be investigated. For example, a radical scavenger antioxidant will hypothetically have a bigger impact on the induction time than a metal chelating antioxidant. The latter will most likely primarily limit the growth rate, since it reduces the amount of pro-oxidative transition metal in the system.

### 3.2. Selection and Training of Sigmoidal Models

For model selection, we excluded LOOH-concentrations above 40 mmol/kg, which was slightly above the estimated CC_LOOH_ of 38 mmol/kg, since these clearly would not fit the sigmoidal function. Both the Gompertz (G, Equation (1)) and normal Foubert (F4, Equation (2)) functions were fitted to hydroperoxide generation curves acquired for a wide range of mayonnaise formulations stored at 50 °C (238 curves). The boundaries of the parameters are described in Appendix A, with most notably the fitting boundaries for the asymptote (A) for both functions. Here, the data was fitted only with an A variation from 50 to 70 mmol/kg to ensure all curve-fittings did not reach their limit before the estimated CC_LOOH_ of 38 mmol/kg. The G and F4 functions were compared by determining the AIC, BIC, and LILC criteria for each curve. These criteria increasingly compensate for the number of samples in the series (LILC > BIC > AIC). According to all three criteria, the Foubert function fitted the data better than the Gompertz functions (Table 1).

As mentioned before, the Foubert function included a parameter n that accounts for the asymmetry of the system. We hypothesized that this asymmetry was similar for all tested systems as they underwent the same ASLT. To estimate a good n value, the averages of n in a variety of sample series were taken and tested throughout the whole dataset. It was found that for *n* = 16.7 (Appendix A) the model matches the ASLT conditions, i.e., shelf-life tests performed at 50 °C. By setting the n to a fixed number, the number of independent variables in the Foubert function were reduced from four to three. This does again impact the assessment on the quality of the fit. Hence, the criterion comparisons were repeated including the reduced Foubert function (F3) with *n* = 16.7. According to the AIC and BIC, F3 was the best fit in over half of the curves, with F4 a close second (Table 2). For the LILC criterion, F4 and F3 did not differ. According to all criteria, the Gompertz function was the poorest fitting function. To estimate whether the difference between the two Foubert functions was significant, an F-test was performed. Respectively, for p = 0.05 and p = 0.01, in 68% and 77% of the cases, the F4-function was not significantly better in fitting the data than the F3-function. We were therefore confident to use the reduced Foubert (F3) in the following prediction and validation studies (Equation (2), with *n* = 16.7).

### 3.3. Aldehyde Onset Prediction

Now that a function had been selected to quantitatively describe the LOOH-generation under accelerated conditions, we pursued the prediction of the time of the aldehyde onset. This prediction model was set up using datasets that comprised both the LOOH curves and aldehydes curves that showed a clear onset. These curves could be primarily grouped in three datasets (A, B, C) according to oil levels (65%, 75%, and 78%, more detail in Appendix A). To establish a robust prediction model that allows variation in base formulations, the three different groups were treated as one dataset. The aldehyde onset for each curve was objectively determined by using the cross-point determination method (see Section 2.4.3). In our three different formulation groups, the CC_LOOH_, the hydroperoxide concentration at which the second acceleration started, differed more than 10 mmol/kg, from 38 to 50 mmol/kg. Figure 2A shows the time difference (Δt_CC-AO_) between the moment the CC_LOOH_ was reached (t_CC_), and the aldehyde onset (t_AO_). The Δt_CC-AO_ values differed between, and within the formulation groups. Hence, there was no constant delay between reaching CC_LOOH_ and the onset of aldehyde generation.

Therefore, we opted for a more robust and empirical approach. Instead of looking at the time difference, we directly looked at the time at which a CC_LOOH_ was reached. Instead of pinpointing the exact CC_LOOH_, the value on the lower end (38 mmol/kg) of the observed CC_LOOH_ (38–50 mmol/kg) was used and the time at which this CC_LOOH_ of 38 mmol/kg was reached was correlated with the aldehyde onset. Linear regression showed that the aldehyde onset can be predicted from the time at which CC_LOOH_ of 38 mmol/kg is reached (Figure 2B and Appendix A). This approach implies a longer delay between the aldehyde onset and CCLOOH time if the formulation is more stable, i.e., due to the presence of more antioxidants or less pro-oxidants.

### 3.4. Model Validation

Lastly, we validated the Foubert (F3) model and its predictive performance for the onset of aldehydes. A dataset with mayonnaises containing both primary and secondary antioxidants (AOs) was designed and the oxidation under ASLT conditions was tracked until the aldehyde onset was reached for most of the formulations. We included three duplications to assess the reproducibility of the method. The AOs that were chosen were GA as a primary AO-system, and EDTA as a secondary AO-system. In this setup, both AOs were dosed in three different levels, in every combination. Since GA and EDTA have different AO-efficiencies, they were dosed in different increments (GA per 75 ppm, EDTA per 10 ppm) to obtain a more comparable effect on overall oxidation. The dataset covered a wide range of LOOH curvatures (Figure 3 and Appendix A). This confirmed that the AOs were dosed in a working range allowing to probe adequately the contribution of the different AO systems. For all formulations, the curves reached 40 mmol/kg of LOOH, so both sigmoidal progression and the secondary acceleration stage were covered. Variation between duplicate curves was very low, both on the formulation level and sample level (both <5%). For the aldehyde curves (Figure 3 and Appendix A), the differentiation between the samples was also present. All, except the 75 ppm EDTA-formulations, reached a clear onset point and could therefore be used for the validation of the aldehyde-onset-prediction model (Figure 4, open bars).

In the previous section, we proposed a simple linear regression (y = 1.6x) to predict the onset of aldehydes on the basis of the time at which CC_LOOH_ was reached using the F3 model. Two different approaches of describing the LOOH-data were explored to predict the aldehyde onset. First, as with the original data, all LOOH concentrations below 40 mmol/kg were used to fit the model (Equation (2)). From this model, the storage time after which the function approached the proposed CC_LOOH_ (38 mmol/kg) was used to predict the aldehyde onset (Figure 4, solid bars). In all eleven formulations, no significant difference (*p* < 0.05) was observed between the predicted and measured aldehyde onsets. In the second approach, only the data from the first week (7 days with daily sampling) were used to build the LOOH-model (Figure 4, dashed bars). For nine out of the eleven formulations, the predicted aldehyde onset was not significantly different from the measured aldehyde onset (*p* < 0.05). However, in the majority of the models, the standard error was visibly bigger than when using the full data. In two cases, the predicted onsets were significantly different than the measured ones (*p* < 0.05). Both these formulations consisted of 150 ppm GA, which hampers LOOH generation in the early stages of the process and did not show a clear sigmoidal curve in the first 7 days. This lack of early data points limited the accuracy of the F3 function. For both approaches, the duplicate formulations were not significantly different from one another (*p* < 0.01).

In summary, we have shown that the time at which a CC_LOOH_ of 38 mmol/kg is reached can be used to predict the aldehyde onset. This prediction model allows us to accurately predict the onset of aldehydes by using hydroperoxide data measured in the first 7 days. The accuracy of the prediction is limited in systems with a high load of AO. In those cases, it would be advised to measure beyond the 7 days mark. This means that early LOOH measurements can be used to predict compromised shelf-life instead of using analytical time-consuming determination of the onset of aldehydes, such as hexanal measurements. The correlation between CC_LOOH_ and aldehyde onsets is in line with our hypothesis that secondary oxidation is accelerated once micelles are formed. We note that correlation does not establish causality.

### 3.5. Chemical Interpretation of the Model Parameters K and f0

The same dataset was used for the chemical interpretation of the two main describing parameters of the model, K and f0 (overview in Appendix A). The f0 (measure for induction delay) and K (growth rate), were respectively hypothesized to be linked to primary (radical scavenging) and secondary (metal chelating) antioxidant effects. Two-way ANCOVA (Equations (5) and (6)) was performed to identify the significant AO factors to f0 and K (Table 2). No significant (*p* < 0.05) interaction-effect between the two AOs on both response parameters was observed, which meant that GA and EDTA did not significantly influence each other in their AO-mechanisms. We observed that GA and EDTA had significant contributions on both f0 and K (p-values in Table 2). The absolute values of the ANCOVA regression model ([ppb]^−1^) represented the relative effects that GA and EDTA had on these two response parameters.

For f0, the effect of EDTA and GA per ppb was not significantly different from each other (*p* < 0.05). This meant that on the weight basis, the effects of EDTA and GA were similar and that f0 is not only sensitive for radical scavenging effects (GA) but also for the chelation effect (EDTA) in the early stages of lipid oxidation. This latter observation is in line with recent work that describes that in the early state of oxidation, LOOH can be generated via LH radicals as well as iron redox cycling [12,25].

For K, a much stronger effect was observed for EDTA, supporting our hypothesis that the main effect of K can be explained by secondary AO-effects. Increasing the amount of EDTA lowers the effective pro-oxidant iron, thereby slowing the reaction. The small, but significant effect of GA on K is in line with this compound being a weak chelator for iron [32]. Our observations suggest that our modelling approach can be used to indicate the primary mode of action (primary vs. secondary) of an AO that is added to the emulsion system.

## 4. Conclusions

A sigmoidal Foubert-adapted model can be used to describe the initial phase of hydroperoxide generation, measured by NMR, under an accelerated shelf-life condition (50 °C). The point at which this function reaches the proposed CC_LOOH_ of 38 mmol/kg can be used as a predictor for the onset of aldehyde generation. This prediction can be applied in different mayonnaise formulations. In most cases, LOOH measurements in the first week of shelf-life are sufficient to accurately predict the aldehyde onset. They could therefore provide an early and quantitative assessment of compromised shelf-life stability. Furthermore, the parameters that describe the Foubert function, i.e., the induction delay (f0) and the growth rate (K), can be used to discriminate primary and secondary anti-oxidative mechanisms.

## Figures and Tables

**Figure 1 antioxidants-10-00287-f001:**
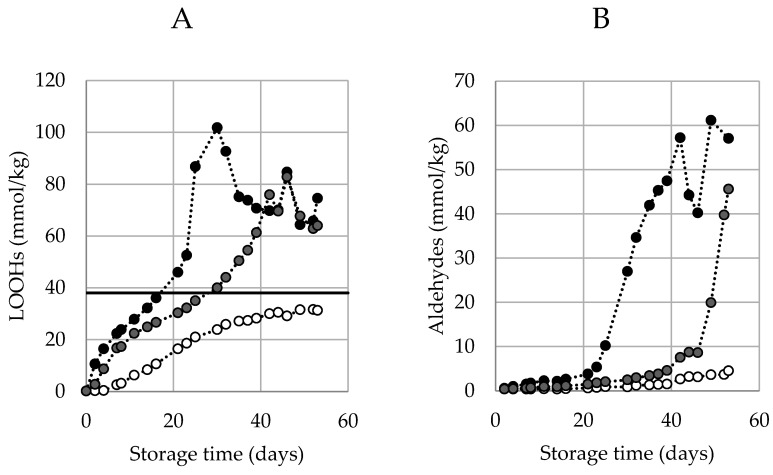
Typical profile of hydroperoxides (LOOHs) (**A**) and aldehyde (**B**) generation over time under accelerated shelf-life conditions (50 °C) measured by proton Nuclear Magnetic Resonance (^1^H NMR). Three different formulations are displayed: no ethylenediaminetetraacetic acid (EDTA) (black), EDTA-alternative (grey), EDTA (white). In A, a line is drawn at 38 mmol/kg, the estimated critical concentration of LOOHs (CC_LOOH_), after which the secondary acceleration is engaged. Lines are solely drawn with the purpose of guiding the eye (no fit).

**Figure 2 antioxidants-10-00287-f002:**
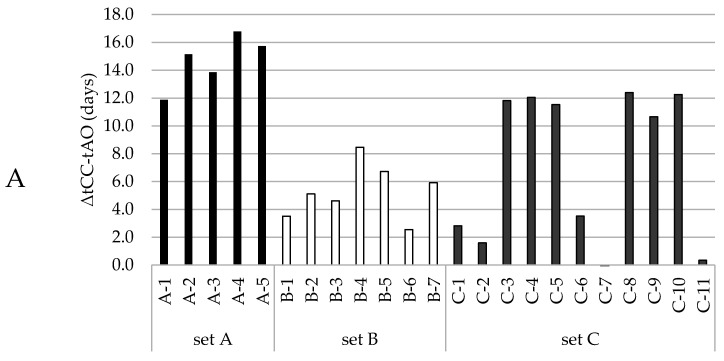
Dependencies between the time after which the CC_LOOH_ was reached and the onset of the corresponding aldehyde curve. Different colors represent the different oil contents in the mayonnaises (set A (65% rapeseed (RP)) black; set B (75% RP) white; set C (78% RP) gray). (**A**) Time differences between the moment CC_LOOH_ was reached and the aldehyde onset (Δt_CC-AO_), where every bar represents a formulation within the mayonnaise set. (**B**) Linear regression model with the time at CC_LOOH_ as regressor and aldehyde onset as predictor, where every circle represents a formulation within the mayonnaise sample set.

**Figure 3 antioxidants-10-00287-f003:**
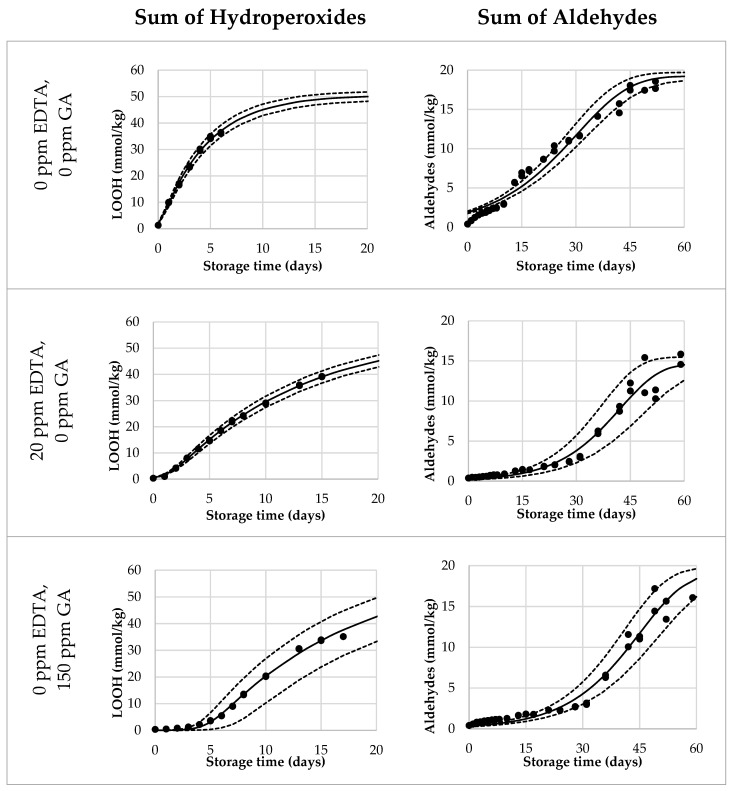
Selection of reduced Foubert (F3) regression results. The dots are the measured data points by ^1^H NMR, the solid line corresponds to the F3 function, and the dashed lines represent the 95% confidence interval (CI). The left graphs represent the LOOH-data up to 40 mmol/kg, fitted with the F3 function with *n* = 16.7. The right graphs represent the aldehyde-data, fitted with the F3 function, with *n* = 1.1 to be used for the cross-point determination.

**Figure 4 antioxidants-10-00287-f004:**
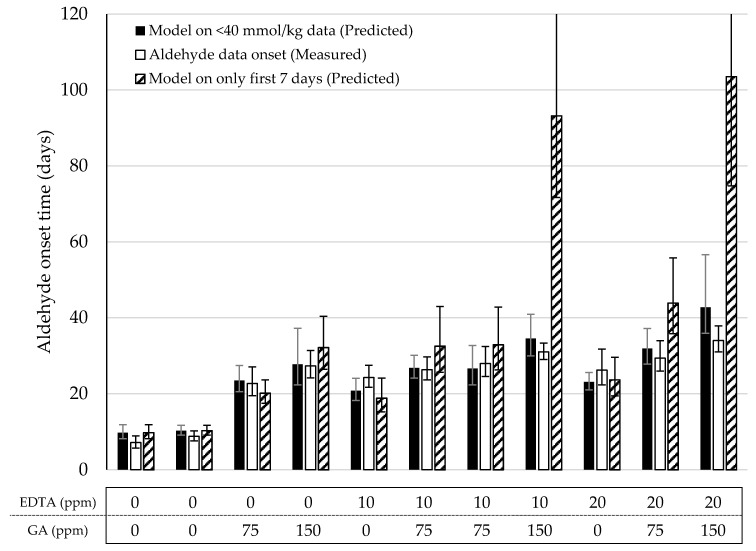
Overview of the predicted and measured aldehyde onset times of the mayonnaises from the model validation under Accelerated Shelf-Life Test (ASLT) conditions (50 °C). The predicted aldehyde onset based on the reduced Foubert (F3) function and the prediction model (Figure 2B) was determined using all data <40 mmol/kg (closed bars) and only the first seven days (dashed bars). Open bars are the onset values based on the cross-point determination obtained from the measured sum of aldehydes. All data was measured with ^1^H NMR.

**Table 1 antioxidants-10-00287-t001:** Comparison of information criteria Aikake Information Criterion (AIC), Bayesian Information Criterion (BIC), and Law of Iterated Logarithm Criterion (LILC) for the models G, F3, and F4. The displayed numbers represent the number of curves (within the 238 curves tested) that presented the lowest criterion-value for each model.

	Gompertz (G)	Normal Foubert (F4)	Reduced Foubert (F3)	Order Model Qualities
AIC	18	98	122	F3 > F4 >> G
BIC	19	92	127	F3 > F4 >> G
LILC	17	111	110	F3 ≈ F4 >> G

**Table 2 antioxidants-10-00287-t002:** Two ways analysis of covariance (ANCOVA) results, after log-transformation. The reported values in this table represent the regression coefficients of the ANCOVA model [ppb]^−1^, which can be interpreted as the effect size of each antioxidant (AO) on the response parameters f0 and K.

	f0 [ppb]^−1^	K [ppb]^−1^
EDTA	−64.4 ± 40.3 **	−50.0 ± 20.3 ***
Gallic Acid	−33.2 ± 5.4 ***	−7.2 ± 2.7 ***

** *p* < 0.01, *** *p* < 0.001.

## Data Availability

Not applicable.

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
