# Peer review of "Quantitative and Predictive Modelling of Lipid Oxidation in Mayonnaise"

_antioxidants, 2021, doi:10.3390/antiox10020287_

Round 1
Reviewer 1 Report
Food with high amounts of fats, especially unsaturated fatty acids, are prone to lipid oxidation process. Various antioxidants are used in the food products to increase its shelf life. Accelerated stability tests or long-term tests are usually used to assess the oxidative stability of various formulations. In order to further refine research on the process of fat degradation in food, it is advisable to develop mathematical models describing such changes.
The work before acceptance should be significantly improved (corrected), especially the introductory and methodological part.
Detailed comments:
Regarding Keywords
Keywords should not be the same as in the title of the work.
Regarding Introduction
- Lines 36-.: “To assess the oxidative stability of food, which under consumer relevant conditions are stable for many months, the food industry applies accelerated shelf life tests, where the oxidation reactions are stimulated to enable faster assessment (…)”- no information about other accelerated tests, e.g. Rancimat test, differential scanning calorimetry (DSC). In addition, the introduction lacks information on research into antioxidants in food. The introduction needs to be completed in this regard. Examples of works on this subject: Biomolecules 2019, 9, 858; doi:10.3390/biom9120858 , https://doi.org/10.1007/s10973-018-7301-0
- Lines 51-53: ”In this work, lipid oxidation in food emulsions is accelerated by storage at elevated temperature (50 °C), where we assume that inter-droplet transport and diffusion mechanisms [9, 10]” - does this information concern the research of the authors of the submitted work or is it information from the literature? The introduction should only contain information found in the available scientific literature.
-Lines 53-55: „We hypothesized that the main mechanisms at play are the known free radical mediated reactions and we also attribute a colloidal role to reverse micelle formation [11, 12]”.
Do the authors mean their published work or other authors?
-Lines 72-75: “In recent years, new techniques [21, 22] have been developed to complement the existing methods [23, 24] that allow…” - These new techniques and existing methods should be mentioned.
Regarding Materials
In this part of the work, the authors described the raw materials rather than the test samples. The number and type of samples that have been tested should be indicated.
Regarding “Sample preparation” – what kind of homogenizer was used for the production of mayonnaise?
Lines 102-103: “Prior to the emulsification, antioxidants (if dosed) were added to the water phase.” – What antioxidants and in what amounts were added? What does "if dosed" mean? I propose to present the recipe composition of the tested samples in the form of a table.
Regarding “NMR measurements” – What type of centrifuge was used? Information should be completed (e.g. type, manufacturer).
Regarding Figures
The work should only include own, graphic studies of the results (not as in Figure S5). Moreover, all abbreviations used in the work (abstract, in the text, figures and tables) should be explained.
Figure: „(Figure&D)” -??
Author Response
Many thanks for reviewing our manuscript. Please find my responses in the attachment.

Reviewer 2 Report
The authors developed a predictive model for volatile off-flavor generation to enable earlier assessment of compromised shelf-life. The mode showed the early and quantitative assessment of compromised shelf-life stability. Several revisions are requested to further improve the superiority of this study and model.
Comment 1
The authors used 1H NMR for the analysis of LOOH, and several molecular species can be analyzed. It was assumed that the total amount of LOOH was used in this study, but were there any differences in the results obtained for different molecular species?
For example, the molecular species produced by auto-oxidation is different from the LOOH species produced by singlet oxygen-oxidation. The differences in the results of the analysis for each molecular species should be shown and discussed.
Comment 2
The number of samples was unclear for some of the results presented in this study.
The number of samples should be indicated, and if the experiment was conducted with n < 3, it should be retested with n ≧ 3.
Author Response

(The authors gave the same response as above.)

Round 2
Reviewer 1 Report
The manuscript is carefully prepared and is suitable for publication in Antioxidants in its current form.
Reviewer 2 Report
The authors have responded appropriately to the reviewer's comments and this manuscript should be accepted.